# Novel Somatic Genetic Variants as Predictors of Resistance to EGFR-Targeted Therapies in Metastatic Colorectal Cancer Patients

**DOI:** 10.3390/cancers12082245

**Published:** 2020-08-11

**Authors:** Pau Riera, Benjamín Rodríguez-Santiago, Adriana Lasa, Lidia Gonzalez-Quereda, Berta Martín, Juliana Salazar, Ana Sebio, Anna C. Virgili, Jordi Minguillón, Cristina Camps, Jordi Surrallés, David Páez

**Affiliations:** 1Genetics Department, Hospital de la Santa Creu i Sant Pau, 08041 Barcelona, Spain; brodriguezs@santpau.cat (B.R.-S.); alasa@santpau.cat (A.L.); lgonzalezq@santpau.cat (L.G.-Q.); jminguillonp@santpau.cat (J.M.); ccampsf@santpau.cat (C.C.); jsurralles@santpau.cat (J.S.); 2Pharmacy Department, Hospital de la Santa Creu i Sant Pau, 08041 Barcelona, Spain; 3Faculty of Pharmacy and Food Sciences, Universitat de Barcelona (UB), 08028 Barcelona, Spain; 4Join Research Unit on Genomic Medicine UAB-IR Sant Pau, Biomedical Research Institute, Hospital de la Santa Creu i Sant Pau, 08041 Barcelona, Spain; 5U705 and U745, ISCIII Center for Biomedical Research on Rare Diseases (CIBERER), 08041 Barcelona, Spain; 6Medical Oncology Department, Hospital de la Santa Creu i Sant Pau, 08041 Barcelona, Spain; bmartinc@santpau.cat (B.M.); asebio@santpau.cat (A.S.); avirgili@santpau.cat (A.C.V.); 7Translational Medical Oncology Laboratory, Institut de Recerca Biomèdica Sant Pau (IIB-Sant Pau), 08041 Barcelona, Spain; jsalazar@santpau.cat; 8Department of Genetics and Microbiology, Universitat Autònoma de Barcelona, 08193 Bellaterra, Spain

**Keywords:** genetic variants, predictive biomarkers, anti-EGFR monoclonal antibodies, colorectal cancer, case-control study

## Abstract

Background: About 40% of *RAS/BRAF* wild-type metastatic colorectal cancer (mCRC) patients undergoing anti-EGFR-based therapy have poor outcomes. Treatment failure is not only associated with poorer prognosis but higher healthcare costs. Our aim was to identify novel somatic genetic variants in the primary tumor and assess their effect on anti-EGFR response. Patients and Methods: Tumor (somatic) and blood (germline) DNA samples were obtained from two well-defined cohorts of mCRC patients, those sensitive and those resistant to EGFR blockade. Genetic variant screening of 43 EGFR-related genes was performed using targeted next-generation sequencing (NGS). Relevant clinical data were collected through chart review to assess genetic results. Results: Among 61 patients, 38 were sensitive and 23 were resistant to treatment. We identified eight somatic variants that predicted non-response. Three were located in insulin-related genes (I668N and E1218K in *IGF1R*, T1156M in *IRS2*) and three in genes belonging to the LRIG family (T152T in *LRIG1*, S697L in *LRIG2* and V812M in *LRIG3*). The remaining two variants were found in *NRAS* (G115Efs*46) and *PDGFRA* (T301T). We did not identify any somatic variants related to good response. Conclusions: This study provides evidence that novel somatic genetic variants along the EGFR-triggered pathway could modulate the response to anti-EGFR drugs in mCRC patients. It also highlights the influence of insulin-related genes and *LRIG* genes on anti-EGFR efficacy. Our findings could help characterize patients who are resistant to anti-EGFR blockade despite harboring *RAS/BRAF* wild-type tumors.

## 1. Introduction

The epidermal growth factor receptor (EGFR), usually overexpressed in colorectal cancer (CRC), plays a pivotal role in tumor growth and progression [1,2]. Multiple proteins are involved in the EGFR signaling pathway, including other receptors, ligands, intracellular downstream effectors, and regulators [2,3,4]. In the metastatic setting, anti-EGFR targeted antibodies, cetuximab and panitumumab, are commonly used, but response rates are variable [5,6]. Several somatic mutations along the EGFR-triggered pathway, such as activating *RAS* mutations and the *BRAF* V600E mutation, are validated predictors of primary resistance to anti-EGFR-based therapies [7,8,9,10,11,12]. Other promising biomarkers of non-response are *PIK3CA* or *PTEN* mutations, although the level of evidence is lower [13,14]. In addition, some studies have observed that right-sided and mesenchymal tumors show worse outcomes to EGFR-targeted therapies regardless of *RAS* mutation status [15,16,17,18]. As about 40% of *RAS*-wild-type patients undergoing anti-EGFR therapy do not benefit from this treatment [19,20], we hypothesized that other mutations in the EGFR pathway could act as additional mechanisms of resistance to EGFR blockade.

Next-generation sequencing (NGS) technologies have revolutionized research in cancer genomics. NGS allows for the simultaneous analysis from several genes to complete genomes with higher sensitivity and cost-effectiveness than the previously used Sanger sequencing methods. In addition, its high sensitivity has detected somatic variants in the tumor at a low allelic fraction.

In this study we used NGS technology to analyze the exons and intron boundaries of 43 EGFR pathway-related genes. We genotyped both germline and tumor DNA samples to optimize the identification of somatic genetic variants. The main objective was to identify novel genetic variants in two cohorts of extreme responder patients with *RAS* wild-type metastatic CRC (mCRC). Extreme responders were either primary resistant or highly sensitive to anti-EGFR therapy. We also aimed to increase knowledge of EGFR-related genes as this could lead to the identification of new biomarkers of anti-EGFR response and promising therapeutic targets for mCRC.

## 2. Patients and Methods

### 2.1. Patient Population

We conducted this case-control study with mCRC patients from the Hospital de la Santa Creu i Sant Pau (HSCSP, Barcelona, Spain). We retrospectively analyzed patients who underwent any anti-EGFR-containing regimen between 2012 and 2017. All tumor samples had been previously classified as *RAS* wild-type using the therascreen *KRAS* test (Qiagen, Hilden, Germany) or the TruSight Tumor 15 panel (Illumina, San Diego, CA, USA).

Two well-defined cohorts of patients were studied, those sensitive (control group) and those resistant (case group) to EGFR blockade. Response to the anti-EGFR-based treatment was determined by total body CT scan. The first CT scan reassessment was performed 2–3 months after treatment. Patients showing disease progression in the first CT scan were classified as resistant. Conversely, those achieving a complete or partial response at this time point, or disease stabilization lasting at least 6 months were considered sensitive. Response to anti-EGFR treatment was assessed according to RECIST (Response Evaluation Criteria in Solid Tumors) v1.1 [21]. An Eastern cooperative oncology group (ECOG) performance status ≤ 2 and age ≥ 18 was required for inclusion in the study. We excluded patients for whom tumor DNA was not available.

Clinical data collected from hospital records included gender, age, performance status (PS) according to the ECOG scale, smoking habit, primary tumor location, number of metastatic sites, time to metastases, resection of the primary tumor, previous lines of chemotherapy, type of anti-EGFR administered, and concomitant chemotherapy. Formalin-fixed paraffin-embedded (FFPE) primary tumor tissues were available from all patients. Germline DNA was also available for 92% of the patients. The study was approved by the Institutional Ethics Committee at HSCSP (ethical code: 22/2012) and all study participants gave written informed consent.

### 2.2. Gene Selection and Primer Design

Two custom panels were created, one for blood samples and the other for tumor samples. Custom amplicon oligonucleotides were designed for each region of interest following the manufacturer’s instructions (Illumina, San Diego, CA, USA). As FFPE tumor DNA is more degraded and fragmented than germline DNA, amplicons of less length are needed to achieve good quality sequencing reads. The panel for germline DNA therefore contained 771 amplicons with an average size of 250–300 base pairs (bp) whereas the panel for tumor DNA contained 1124 amplicons of ~175 bp. Both panels included the 43 candidate genes related to the EGFR pathway and had an expected coverage >99% for all the coding regions (~200,000 bp). These genes mainly encoded receptors, ligands, intracellular downstream effectors or proteins involved in EGFR turnover. We included the most relevant genes of the pathway and also those related to anti-EGFR response in previous studies [3,22,23]. Table 1 provides information about the selected genes and the function of their corresponding encoded proteins.

### 2.3. Isolation and Quantification of DNA

Germline DNA was automatically extracted from peripheral whole-blood samples (Autopure, Qiagen, Hilden, Germany). Tumor DNA was extracted from primary tumor tissue samples using the GeneRead DNA FFPE Kit (Qiagen, Hilden, Germany). This kit purifies tumor DNA by removing artificial C > T mutations. DNA concentrations were measured using Qubit™ dsDNA HS Assay Kits with the Qubit 3.0 Fluorometer (Invitrogen, Carlsbad, CA, USA).

### 2.4. Library Preparation, Sequencing Runs, and NGS Analysis

Library preparation was carried out following the manufacturer’s instructions (TruSeq Custom Amplicon Low Input Library Prep, Illumina, San Diego, CA, USA). We used ~15 ng of germline DNA or ~100 ng of tumor DNA, due to the low quality of DNA from FFPE samples. Library sizes were determined using QIAxcel DNA Screening Gel Cartridge on QIAxcel capillary electrophoresis system (Qiagen GmbH, Hilden, Germany). Library concentrations were measured with Qubit 3.0 Fluorometer (Invitrogen, Carlsbad, CA, USA). All libraries were subsequently diluted and pooled equimolarly.

Germline DNA sequencing was performed on a MiSeq platform (Illumina) to obtain 150-bp paired-end reads. Samples were sequenced using the MiSeq v2-300 Reagent Kit. To achieve higher coverage, tumor DNA sequencing was performed on a NextSeq 500 platform (Illumina), also obtaining 150-bp paired-end reads. Samples were sequenced using the NextSeq Mid v2-300 Reagent Kit.

Sequence reads were aligned against the human reference genome (version GRCh37) using the Burrows-Wheeler Aligner (BWA, version 0.7.12) [24]. Single nucleotide and indel variants were called by means of the Mutect2 tool (version 4.0.12.0) that can use both input tumor and germline data. It can also manage high coverage data from tumor sequences [25]. Alignment and calling were performed following software development best practices (Broad Institute, Cambridge, MA, USA). After alignment, a panel of normal variation (PoN) obtained from germline DNA sequencing data was used to exclude rare germline variants and individual-specific artifacts. The number of somatic variants per patient was calculated by filtering tumor DNA sequencing data with the PoN. We excluded variants located in intronic, intergenic or UTR regions, and also polymorphisms (GnomAD allele frequency >0.001). Results were inspected using the Integrative Genomics Viewer (IGV). Resistant patients whose tumors harbored a known non-response mutation (in *KRAS*, *NRAS* or *BRAF V600E*) of over 5% of mutated clones were excluded. The presence of these non-response mutations was confirmed by a second technique (Sanger sequencing or the TruSight Tumor 15 panel) in some patients. Finally, only those variants that were over 1% of mutant clones in all patients harboring them were kept for further examination. COSMIC cancer database v91 was used to assess whether the candidate somatic variants identified had been previously described [26]. Additional functional annotation of variants was performed using ANNOVAR [27]. Variant analyses and interpretation were performed using Alamut^®^ Visual v2.15 software (SOPHiA GENETICS, Boston, MA, USA) and the Cancer Genome Interpreter platform (Institute for Research in Biomedicine, Barcelona, Spain), which is publicly available at http://www.cancergenomeinterpreter.org [28].

### 2.5. Statistical Analyses

We used the chi-square test to compare the baseline clinical characteristics between the two cohorts of patients and Fisher’s exact test to compare the prevalence of alterations between sensitive and resistant patients. The Mann-Whitney test was carried out to compare the number of somatic genetic variants between sensitive and resistant patients. All statistical tests were performed using R software (version 3.3.2., https://cran.r-project.org/bin/windows/base/old/3.3.2/) and IBM SPSS^®^ statistics software (version 25, https://www.ibm.com/support/pages/downloading-ibm-spss-statistics-25). A 95% confidence level was set for all tests of significance.

## 3. Results

### 3.1. Patient Population

A total of 168 mCRC patients were treated with anti-EGFR-containing regimens between 2012 and 2017. Sixty-one of the patients (38 sensitive and 23 resistant to EGFR blockade) were included in the study, as they met the inclusion criteria and quality of tumor DNA was good (Figure 1). They were all diagnosed with stage IV CRC (37.7% metachronous, 50.8% ≥ 2 metastatic locations). Regarding treatment, 33 patients received cetuximab and 28 received a panitumumab-containing regimen, with no significant differences between the two cohorts (*p* = 0.814). There were more females and more patients with worse ECOG PS in the non-responder cohort (*p* < 0.001 and *p* = 0.013, respectively). No differences were observed between the two cohorts in respect to primary tumor location (*p* = 0.634). A higher percentage of patients in the sensitive group received the anti-EGFR-containing regimen as first-line treatment (47.4% in the sensitive group vs. 26.1% in the resistant group) and presented synchronous metastases (71.1% in the sensitive group vs. 47.8% in the resistant group). Baseline clinical features are described in Table 2.

### 3.2. Genetic Analyses

The mean target coverage for tumor samples was 3600, achieving 100× or greater coverage for 88% of the bases. For germline samples, the mean target coverage was 640, achieving 30× or greater coverage for 87% of the bases. As shown in Figure 2, we identified 46,512 somatic genetic variants. We found a mean of 707 variants per sample in sensitive patients and 854 variants per sample in resistant patients (*p* = 0.19).

### 3.3. KRAS, NRAS, BRAF, and PIK3CA Assessment and Patient Selection

Five anti-EGFR resistant patients were finally excluded from the analyses because a validated mutation of non-response (in *RAS* or *BRAF^V600E^*) was detected at an allelic fraction over 5%. One of the five patients (P11) had the *KRAS* G12C mutation. The remaining four patients (P3, P39, P55, and P64), all with right-sided tumors, presented the *BRAF* V600E mutation. In one patient (P64) this mutation had not been previously detected by Sanger. Patient P55 presented the mutations *BRAF* V600E and *NRAS* G13D at a frequency over 5% for both mutations. As for patient P39, *BRAF* V600E and *KRAS* Q61L mutations coexisted, although the allelic fraction for *KRAS* Q61L was only 0.8%. No patients harbored somatic mutations in both *KRAS* and *NRAS* genes. One sensitive patient (P28) who underwent FOLFIRI plus panitumumab as a second-line treatment presented the *KRAS* A146V mutation at an allelic fraction of 5.0%. This mutation had not been identified previously. Table 3 and Appendix A show all the somatic mutations found in these genes and their allelic fractions.

We also analyzed *PIK3CA* mutations and *BRAF* mutations other than V600E. Two patients, one resistant (P63) and one sensitive (P66), presented the *PIK3CA* E545K mutation, with an allelic fraction of 10.1% and 6.1%, respectively. In addition, two *BRAF* mutations previously related to favorable prognosis (D594N and G466A) [29,30,31,32] were identified at around 20% in two sensitive patients (P45 and P59). Table 3 shows all the findings concerning *BRAF* and *PIK3CA* assessment.

### 3.4. Identification of Novel Genetic Variants Related to Anti-EGFR Response

We identified eight potential somatic variants of non-response at a frequency of over 1% in 12 out of 18 (66.7%) non-responder patients (Table 4). In 8 cases, two or more of these variants coexisted. Mutant allele fractions differed substantially among the patients (Appendix A). No potential resistance variants were found in 6 non-responders. Additionally, no variants of good response were identified.

Three of the eight variants detected were missense variants located in insulin-related genes, such as *IGF1R* I668N and E1218K or *IRS2* T1156M. Three others were found in genes belonging to the LRIG family: *LRIG1* (T152T), *LRIG2* (S697L), and *LRIG3* (V812M). The remaining two variants were found in *NRAS* (G115Efs*46) and *PDGFRA* (T301T). All variants were non-synonymous, except for *LRIG1* and *PDGFRA* variants. According to Alamut software, these two variants may create a novel cryptic acceptor site identifiable by the splicing complex.

## 4. Discussion

We sought to investigate the existence of novel somatic variants in EGFR-related genes as predictive markers of response to anti-EGFR antibodies. We found eight potential somatic variants that could explain the lack of response to these agents, highlighting the variants in the insulin-related and LRIG family genes. In contrast, we did not find somatic variants related to good response.

Accurate identification of somatic variants is challenging. In the past, Sanger sequencing was the only technique available to detect somatic mutations in tumor samples. Currently, NGS technology is replacing Sanger method as it allows the sequencing of hundreds of genes simultaneously and shows high sensitivity [35]. These advantages have enabled the detection of concomitant mutations in several genes of interest, including low-allele-fraction variants not previously found by Sanger sequencing. Mutant allele fractions may influence the response to targeted therapies. In this sense, the mutant allele fraction that determines anti-EGFR response continues to be debated. A major point of discussion is whether the optimal threshold of the *RAS* mutant allele fraction to identify patients likely to benefit from anti-EGFR drugs should be 1% or 5%. We used a threshold of 5% as it has been reported that reducing the threshold to 1% does not improve outcomes [36,37]. We found one sensitive patient (P28) who harbored the A146V *KRAS* mutation at 5%. The good response in this patient could be related to the chemotherapy scheme concomitantly given with the anti-EGFR drug. We also found a novel somatic variant of non-response in the *NRAS* gene. *NRAS* variants routinely tested prior to anti-EGFR initiation are normally missense mutations (≈95%) [7]. Conversely, the new *NRAS* variant we found, G115Efs*46, is a truncating mutation. It consists of a deletion located in exon 4 that leads to a premature stop codon and a truncated protein [26]. The Cancer Genome Interpreter predicts that it is a passenger mutation with a highly deleterious effect, but its role in anti-EGFR resistance should be further explored before a solid conclusion can be reached.

Similarly to *RAS* genes, *BRAF* and *PIK3CA* are driver oncogenes involved in colorectal carcinogenesis. It has been reported that *BRAF* mutations implying a high kinase activity (such as V600E) confer a poor prognosis, whereas those implying a low kinase activity (such as those located in codons 594 and 596) confer a favorable prognosis [29,30,31]. Our results reinforce the differential prognostic role of *BRAF* mutations as we only detected mutations implying a high kinase activity (V600E) in patients who were resistant to treatment, and we only found mutations causing low kinase activity in sensitive patients (D594N and G466A). As for location, several studies show that V600E mutations are more common in right colon cancers [38,39]. Accordingly, we only found V600E mutations in right-sided tumors. In contrast, we found D594N and G466A mutations in left-sided tumors. This differential distribution of *BRAF* mutations may affect the prognosis of left-sided tumors vs. right-sided tumors and define a clinically distinct subtype of CRC with an excellent prognosis. In addition, one resistant patient harbored *NRAS* and *BRAF* mutations over 5%, indicating that they are not always mutually exclusive. Regarding *PIK3CA*, the role of its activating mutations on anti-EGFR response remains under discussion [14,40,41]. In the present study, two patients, one responder and one non-responder, harbored the *PIK3CA* E545K mutation with an allelic fraction over 5%. Our results therefore strengthen the notion that this mutation is not critical for anti-EGFR response.

Our findings suggest some insulin-related genes (*IGF1R* and *IRS2*) have a substantial influence on anti-EGFR outcomes. A growing amount of evidence indicates that the insulin-like growth factor-1 receptor (IGF1R) is frequently overexpressed in CRC and that its activation is related to poorer outcomes [42]. IGF1R is a tyrosine kinase receptor that can also activate the RAS pathway and promote proliferation of cancer cells, resistance to apoptosis, and epithelial-mesenchymal transition [43]. Furthermore, it has been reported that a functional IGF1R receptor is required for EGFR-mediated growth and transformation [44] and that IGF1R expression modulates anti-EGFR efficacy in mCRC patients [45]. Consequently, in the same way that activating mutations in the EGFR receptor potentiate the RAS pathway, IGF1R activating mutations could play a similar role, with worse responses to EGFR-targeted therapies. To our knowledge, this is the first time that the missense *IGF1R* variants identified in our study (I668N and E1218K) have been related to anti-EGFR resistance. The variant E1218K has been previously described in a patient with leiomyosarcoma [33], but the variant I668N has not been reported previously [26]. We also identified a missense variant in the *IRS2* gene (T1156M) associated with a lack of response to anti-EGFR drugs. Interestingly, Bertotti et al. demonstrated that *IRS2* knockdown reduced sensitivity to cetuximab [23]. Truncating variants in this gene could therefore imply a lack of response to this drug. In our study, we found a missense variant in this gene harbored by three non-responders (Table 4). This variant had already been reported in a patient with colon cancer [33]. Like EGFR and IGF1R signaling pathways, the PDGFRA pathway is also involved in cell proliferation and migration [46]. The platelet-derived growth factor receptor A (PDGFRA) is a tyrosine kinase receptor that is frequently mutated in gastrointestinal stromal tumors [47]. In the present study, we found a splicing variant located in *PDGFRA*. Mutations in this gene have been correlated with a lack of response to anti-EGFR agents, although evidence is still scarce. In this line, Bertotti et al. described novel missense mutations located in/near the catalytic domain as mechanisms of anti-EGFR resistance [23].

We also found that *LRIG1-3* genes could play a role in anti-EGFR response. Little is known about the role of somatic variants in these genes in responsiveness to anti-EGFR agents. Several studies have demonstrated that LRIG1 acts as a tumor suppressor by down-regulating ErbB and Met receptors, including EGFR [48,49,50,51,52]. In contrast, no definitive conclusions concerning the contribution of LRIG2 and LRIG3 to EGFR levels can be drawn as results reported to date are contradictory [48,53,54,55,56]. Gelfo et al. found that a lower LRIG1 expression predicted resistance to cetuximab therapy in CRC xenopatients, an effect that was not observed with LRIG3 [57]. In our study, one somatic variant in each *LRIG* gene was significantly associated with resistance to anti-EGFR blockade. Interestingly, the alteration located in *LRIG1* had already been described in two cancer patients, one with a colon adenocarcinoma and the other with bladder cancer [34]. Conversely, the variants found in *LRIG2* and *LRIG3* have not been previously reported [26].

We wish to emphasize that no somatic variants related to good response were found in our study. This result is not striking as resistant mCRC phenotypes tend to be more heterogeneous than sensitive phenotypes [58]. Our results are in keeping with most papers to date describing mutations of non-response to anti-EGFR antibodies [59,60,61]. Only three of the eight somatic variants identified had been previously described [26], strengthening the extreme phenotype approach as a useful strategy to identify rare alterations. However, our study has some limitations. First, the two cohorts of patients are relatively small. The low number of patients showing an extremely poor response could be the result of good selection by clinicians after assessing *RAS* and *BRAF* mutational status and considering patient’s characteristics. The lack of statistically significant differences in the number of somatic mutations between the two cohorts of patients and between sidedness and drug response could also be attributable to the small sample size. Likewise, the association observed between sex and responsiveness could be a false positive predictive marker. Second, patients were included over a five-year period (2012–2017), during which time scientific knowledge regarding *RAS* and *BRAF* mutations increased significantly. This could explain why some patients with a *BRAF* V600E mutation already detected in the first assessment received an anti-EGFR agent. Larger, prospective and functional studies are needed to confirm the validity of our findings.

Therapies targeting different members and downstream effectors of the EGFR signaling pathway, such as *KRAS* or *BRAF*, have previously been explored [62,63]. However, most patients develop resistance to these agents. Our findings reveal novel mechanisms of resistance to anti-EGFR targeted therapies in mCRC patients that could provide new avenues for therapeutic intervention involving insulin-related and/or LRIG proteins. The utility of IGF1R as a possible therapeutic target has already been evaluated. In non-small cell lung cancer, IGF1R hyperactivity is related to acquired resistance to erlotinib, and simultaneous inhibition of EGFR and IGF1R is effective to prevent and also to overcome erlotinib resistance [64]. In CRC, the simultaneous inhibition of EGFR together with the IGF1R antagonist dalotuzumab has been tested in *KRAS* wild-type mCRC patients showing negative results [65]. Our results, together with the evidences thus far reported, could promote studies assessing insulin-related and also LRIG1-3 proteins as possible therapeutic targets in CRC. The impact of somatic variants in the genes encoding these proteins on anti-EGFR efficacy should also be confirmed.

## 5. Conclusions

In conclusion, this study shows that novel genetic variants along the EGFR-triggered pathway could affect the response to anti-EGFR drugs in mCRC patients. Our findings may help to better identify patients who are resistant to anti-EGFR drugs despite harboring *RAS/BRAF*-wild-type tumors. The eight genetic variants predictive of non-response could help guide clinical decision-making and improve outcomes by tailoring anti-EGFR drugs.

## Figures and Tables

**Figure 1 cancers-12-02245-f001:**
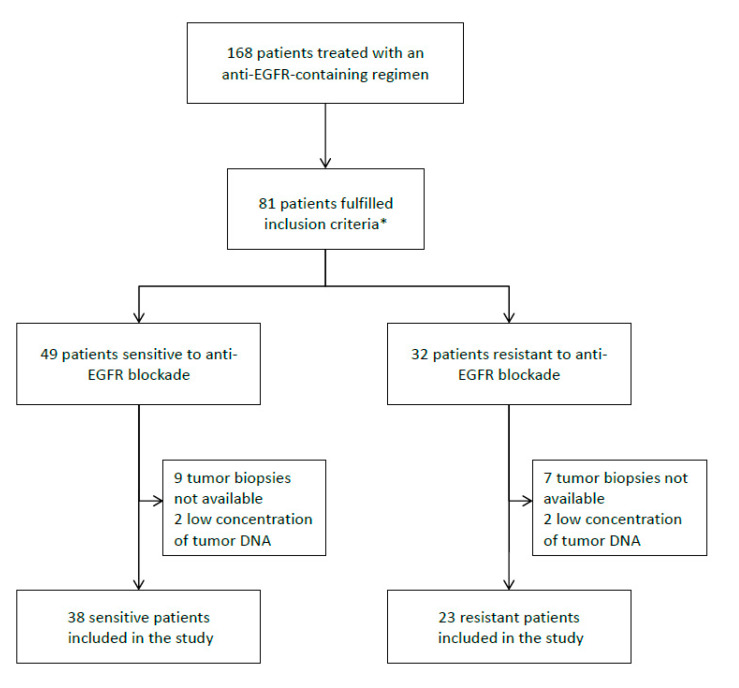
Flow chart of patient selection. * Inclusion criteria: patients ≥ 18 years, with an Eastern cooperative oncology group (ECOG) performance status ≤ 2 and *RAS* wild-type tumors. Patients had to be sensitive (patients achieving complete or partial response at the first CT-scan or disease stabilization lasting at least 6 months) or resistant to anti-EGFR blockade (patients showing disease progression at the first CT-scan).

**Figure 2 cancers-12-02245-f002:**
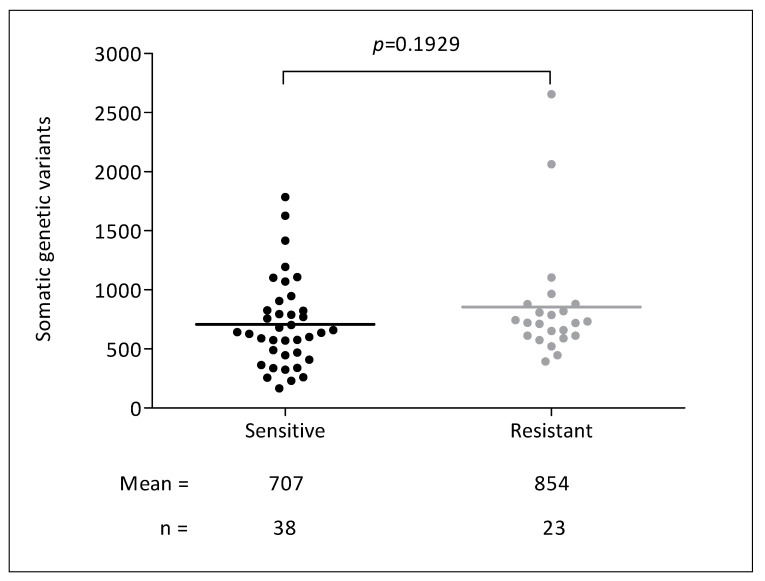
Number of somatic genetic variants and response to anti-EGFR agents.

**Table 1 cancers-12-02245-t001:** Selected genes classified according to the function of their encoding proteins.

Ligands	Receptors	Intracellular Downstream Effectors	Proteins Involved in EGFR Turnover	Others
*AREG*	*EGFR (ERBB1* or *HER1)*	*AKT1*	*AGR2*	*TP53*
*BTC*	*ERBB2 (HER2)*	*BRAF*	*CBL*	*YAP1*
*EGF*	*ERBB3 (HER3)*	*HRAS*	*LRIG1*	
*EPGN*	*ERBB4 (HER4)*	*IRS2*	*LRIG2*	
*EREG*	*FGFR1*	*KRAS*	*LRIG3*	
*HBEGF*	*IGF1R*	*MAP2K1*	*NEDD8*	
*HGF*	*MET*	*NRAS*	*ERRFI1 (RALT* or *MIG6)*	
*IGF1-2*	*PDGFRA*	*PIK3CA*	*SOCS4*	
*NRG1-4*		*PTEN*	*SOCS5*	
*TGFα*			*SPRY2*	

Abbreviations: *AGR2*, Anterior Gradient 2, Protein Disulphide Isomerase Family Member; *AKT1*, AKT Serine/Threonine Kinase 1; *AREG*, Amphiregulin; *BRAF*, B-Raf Proto-Oncogene, Serine/Threonine Kinase; *BTC*, Betacellulin; *CBL*, Cbl Proto-Oncogene; *EGF*, Epidermal Growth Factor; *EGFR*, Epidermal Growth Factor Receptor; *EPGN*, Epithelial Mitogen; *ERBB2*, Erb-B2 Receptor Tyrosine Kinase 2; *ERBB3*, Erb-B3 Receptor Tyrosine Kinase 3; *ERBB4*, Erb-B4 Receptor Tyrosine Kinase 4; *EREG*, Epiregulin; *ERRFI1*, ERBB Receptor Feedback Inhibitor 1; *FGFR1*, Fibroblast Growth Factor Receptor 1; *HBEGF*, Heparin Binding EGF Like Growth Factor; *HGF*, Hepatocyte Growth Factor; *HRAS*, HRas Proto-Oncogene, GTPase; *IGF1*, Insulin Like Growth Factor 1; *IGF2*, Insulin Like Growth Factor 2; *IGF1R*, Insulin Like Growth Factor 1 Receptor; *IRS2*, Insulin Receptor Substrate 2; *KRAS*, KRAS Proto-Oncogene, GTPase; *MAP2K1*, Mitogen-Activated Protein Kinase Kinase 1; *LRIG1*, Leucine Rich Repeats And Immunoglobulin Like Domains 1; *LRIG2*, Leucine Rich Repeats And Immunoglobulin Like Domains 2; *LRIG3*, Leucine Rich Repeats And Immunoglobulin Like Domains 3; *MET*, MET Proto-Oncogene, Receptor Tyrosine Kinase; *NEDD8*, Neural Precursor Cell Expressed, Developmentally Down-Regulated 8; *NRAS*, NRAS Proto-Oncogene, GTPase; *NRG1*, Neuregulin 1; *NRG2*, Neuregulin 2; *NRG3*, Neuregulin 3; *NRG4*, Neuregulin 4; *PDGFRA*, Platelet Derived Growth Factor Receptor Alpha; *PIK3CA*, Phosphatidylinositol-4,5-Bisphosphate 3-Kinase Catalytic Subunit Alpha; *PTEN*, Phosphatase And Tensin Homolog; *SOCS4*, Suppressor Of Cytokine Signaling 4; *SOCS5*, Suppressor Of Cytokine Signaling 5; *SPRY2*, Sprouty RTK Signaling Antagonist 2; *TGFα*, Transforming Growth Factor Alpha; *TP53*, Tumor Protein P53; *YAP1*, Yes Associated Protein 1.

**Table 2 cancers-12-02245-t002:** Baseline patient characteristics.

Characteristic	Study Population (*n* = 61)	Sensitive Patients (*n* = 38)	Resistant Patients (*n* = 23)	*p*-Value
	N (%)	N (%)	N (%)	
Sex				
Male	40 (65.6%)	32 (84%)	8 (34.8%)	**<0.001**
Female	21 (34.4%)	6 (16%)	15 (65.2%)	
Age				
<75	43 (70.5%)	26 (68.4%)	17 (73.9%)	0.649
≥75	18 (29.5%)	12 (31.6%)	6 (26.1%)	
Mean Age		67.7	66.9	
Performance status (ECOG)				
0	31 (50.8%)	24 (63.2%)	7 (30.4%)	**0.013**
1–2	30 (49.2%)	14 (36.8%)	16 (69.6%)	
Smoking habit				
Never smokers	24 (39.3%)	13 (34.2%)	11 (47.8%)	0.291
Current or former smokers	37 (60.7%)	25 (65.8%)	12 (52.2%)	
Tumor side				
Right	24 (39.4%)	15 (39.5%)	9 (39.2%)	
Left	20 (32.7%)	13 (34.2%)	7 (30.4 %)	0.634
Rectal	16 (26.3%)	10 (26.3%)	6 (26.1%)	
Jejunum	1 (1.6%)	0	1 (4.3%)	
Number of metastatic sites				
1	30 (49.2%)	18 (47.4%)	12 (52.3%)	0.716
≥2	31 (50.8%)	20 (52.6%)	11 (47.7%)	
Time to metastases				
Synchronous	38 (62.3%)	27 (71.1%)	11 (47.8%)	0.070
Metachronous	23 (37.7%)	11 (28.9%)	12 (52.2%)	
Primary resected				
Yes	49 (80.3%)	32 (84.2%)	17 (73.9%)	0.327
No	12 (19.7%)	6 (15.8%)	6 (26.1%)	
Previous lines of treatment				
0	24 (39.3%)	18 (47.4%)	6 (26.1%)	
1	32 (52.5%)	17 (44.7%)	15 (65.2%)	0.246
≥2	5 (8.2%)	3 (7.9%)	2 (8.7%)	
Type of anti-EGFR				
Cetuximab	33 (54.1%)	21 (55.3%)	12 (52.2%)	0.814
Panitumumab	28 (45.9%)	17 (44.7%)	11 (47.8%)	
Combination QT				
FOLFOX	18 (29.5%)	14 (36.8 %)	4 (17.4%)	
Irinotecan scheme	40 (65.6%)	23 (60.6%)	17 (73.9%)	0.192
Monotherapy	3 (4.9%)	1 (2.6 %)	2 (8.7%)	
PFS (months)		18.8	4.7	
OS (months)		41.2	17.2	

Abbreviations: ECOG, Eastern Cooperative Oncology Group; OS, overall survival; PFS, progression-free survival. *p*-values below 0.05 are highlighted in bold.

**Table 3 cancers-12-02245-t003:** Assessment of *KRAS*, *NRAS, BRAF,* and *PIK3CA* mutational status by next-generation sequencing.

Treatment Outcome	Gene	Mutation	Patient	% of Mutation	Coverage	Mutational Status Prior to Anti-EGFR Prescription
Resistant	*KRAS*	G12C	P11	45.7%	2316	*KRAS* wild-type
	*KRAS*	Q61L	P39	0.8%	14,637	*KRAS* exon 3 not tested
	*NRAS*	G12S	P51	2.6%	5093	*NRAS* wild-type
	*NRAS*	G13D	P55	8.1%	12,265	*NRAS* not tested
	*NRAS*	G13D	P57	0.5%	41,758	*NRAS* wild-type
	*BRAF*	V600E	P3	24.4%	10,995	*BRAF* V600E mutated
	*BRAF*	V600E	P39	10.6%	7906	*BRAF* V600E mutated
	*BRAF*	V600E	P55	20.8%	22,185	*BRAF* V600E mutated
	*BRAF*	V600E	P64	12.1%	1316	*BRAF* V600E wild-type
	*PIK3CA*	E545K	P63	10.1%	17,051	*PIK3CA* not tested
Sensitive	*KRAS*	A146V	P28	5.0%	2743	*KRAS* wild-type
	*BRAF*	D594N	P45	19.7%	23,584	*BRAF* codon 594 not tested
	*BRAF*	G466A	P59	21.0%	10,234	*BRAF* codon 466 not tested
	*PIK3CA*	E545K	P66	6.1%	12,888	*PIK3CA* not tested

Abbreviations: *BRAF*, B-Raf Proto-Oncogene, Serine/Threonine Kinase; *KRAS*, KRAS Proto-Oncogene, GTPase; *NRAS*, NRAS Proto-Oncogene, GTPase; *PIK3CA*, Phosphatidylinositol-4,5-Bisphosphate 3-Kinase Catalytic Subunit Alpha.

**Table 4 cancers-12-02245-t004:** Genetic variations significantly associated with lack of response to anti-EGFR blockade.

Gene	Genetic Variant	Patients with the Variant	% of the Somatic Variant	Coverage	*p-* Value *	Presence in COSMIC Cancer Database v91
*IGF1R*	NM_001291858.1: c.2003T > A; p.(I668N)	P4	3.5%	8008	0.029	Not described
		P9	4.2%	1745		
		P10	10.6%	3093		
	NM_001291858.1: c.3652G > A; p.(E1218K)	P1	6.3%	2740	0.008	Mutation Id: 6919417 Patient with a leiomyosarcoma (*n* = 1) [33]
		P2	11.8%	1866		
		P9	2.4%	1238		
		P61	1.5%	10,017		
*IRS2*	NM_003749.2: c.3467C > T; p.(T1156M)	P12	7.0%	743	0.029	Mutation Id: 6974893 Patient with colon cancer (*n* = 1) [33]
		P14	5.1%	1093		
		P57	3.4%	11,118		
*LRIG1*	NM_015541: c.456G > A; p.(T152T)	P12	89.3%	196	0.008	Mutation Id: 4005617 Patient with colon cancer (*n* = 1) Patient with bladder cancer (*n* = 1) [34]
		P57	2.7%	12,073		
		P63	7.9%	6106		
		P67	4.0%	3218		
*LRIG2*	NM_014813.2: c.2090C > T; p.(S697L)	P1	2.2%	8802	0.029	Not described
		P10	4.2%	1464		
		P12	7.9%	1794		
*LRIG3*	NM_001136051.2: c.2434G > A; p.(V812M)	P1	1.7%	12,378	0.029	Not described
		P2	4.8%	4477		
		P9	4.6%	2266		
*NRAS*	NM_002524.3: c.344del; p.(G115Efs*46)	P9	1.6%	2273	0.029	Not described
		P10	3.1%	3140		
		P63	2.2%	20,507		
*PDGFRA*	NM_001347828: c.903G > A; p.(T301T)	P52	22.0%	19,078	0.029	Not described
		P63	3.8%	26,774		
		P67	1.2%	55,158		

* *p*-value was obtained by Fisher’s exact test. Abbreviations: *IGF1R*, Insulin-like Growth Factor 1 Receptor; *IRS2*, Insulin Receptor Substrate 2; *LRIG1*, Leucine-rich Repeats and Immunoglobulin-like Domains 1; *LRIG2*, Leucine-rich Repeats and Immunoglobulin-like Domains 2; *LRIG3*, Leucine-rich Repeats and Immunoglobulin-like Domains 3; *NRAS*, NRAS Proto-Oncogene, GTPase; *PDGFRA*, Platelet-derived Growth Factor Receptor Alpha.

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
