# Peer review of "Novel Somatic Genetic Variants as Predictors of Resistance to EGFR-Targeted Therapies in Metastatic Colorectal Cancer Patients"

_cancers, 2020, doi:10.3390/cancers12082245_

Round 1

Reviewer 1 Report

Dear 

It is a valuable study

I suggest to further clarify the methods and widen the discussion

adding a perspective section.

warm regards

Cancers

Title of manuscript: Novel somatic genetic variants as predictors of resistance to EGFR-targeted therapies in metastatic colorectal cancer patients.

I recommend minor revision

COMMENTS

It is a valuable and interesting retrospective study that focused on the novel predictive factors of resistance to EGFR in colon cancer.

Please clarify in the methods section  which is the control group and how many patients were recruited for each group

How was the response to the treatment determined, by CT Total body scan or FDG-PET?

Futher details should be provided regarding peripheral blood sample. Was a peripheral cancer cells count carried out? I think it could be useful to determine peripheral cancer cells before and after tumor progression.

Data regarding smokng habit should be included in the baseline data as well as data regarding staging.

Please clarify the site from which the histology samples were derived whether it is from primary or from metastatic site.

I suggest to broaden the methods and discussion.

Please include limitations of the study since the sample size for each group is small and differences about somatic mutation variants were not significant between the two groups

I suggest to include section on possible perspective deriving from these findings.

Some references eventually should be added to broaden the discussion :

  • Future Sci OA. 2019 May 3;5(5):FSO394. doi: 10.2144/fsoa-2019-0017. Review about the relationship between cigarette smoke-cell cycle and tumor spread
  • HIF in tumor progression associated with EMT pathway which is in turn linked to IGF –R Activation
  • Sci Rep . 2013;3:2560. doi: 10.1038/srep02560 about the anti-EGFR resistance

due to EMT-IGF crosstalk

  • J Surg Oncol . 2000 Oct;75(2):98-102 about the use of novel biomarkers for aggressive colon cancer on peripheral blood
  • Int J Environ Res Public Health. 2014 Aug 25;11(9):8645-60. About clinical parameters associated with KRAS mutation.

Author Response

  1. Please clarify in the methods section which is the control group and how many patients were recruited for each group

As the reviewer suggests, we have now specified the case and control groups in lines 79-80 (Patients and methods). The number of patients recruited for each group is specified in the results section (lines 181-182) and in Figure 1.

  1. How was the response to the treatment determined, by CT Total body scan or FDG-PET?

The response to the treatment was determined by CT total body scan. We have now added this information in lines 80-81, in the Patients and Methods section.

  1. Further details should be provided regarding peripheral blood sample. Was a peripheral cancer cells count carried out? I think it could be useful to determine peripheral cancer cells before and after tumor progression.

We agree with the reviewer that determining the number of peripheral cancer cells before and after tumor progression would be of high interest to monitor disease progression. However, due to the retrospective characteristics of the study, this analysis could not be performed.

  1. Data regarding smoking habit should be included in the baseline data as well as data regarding staging.

Following the reviewer’s comment, we have added this information in Table 2, in lines 89-90 (Patients and methods section) and in lines 182-183 and 190-191 (Results section). Regarding smoking habit, we classified all the patients into two groups (non-smokers vs current/former smokers). No statistically significant differences were found between the two cohorts of patients. As for staging, all patients included in the study had stage IV colorectal cancer when they underwent the anti-EGFR-based regimen. To provide more useful information regarding this aspect, in Table 2 we have also added whether the metastases were synchronous or metachronous. We thank the reviewer for pointing this out.

  1. Please clarify the site from which the histology samples were derived whether it is from primary or from metastatic site.

All tumor samples were derived from the primary tumor. We have now specified this in line 92 (in the section Patient population, in Patients and Methods) and also in line 131 (in the section Isolation and quantification of DNA, in Patients and Methods).

  1. I suggest to broaden the methods and discussion.

Following the reviewer’s suggestion, we have broadened the Patient population sections (in Patients and methods and in Results). We have also added a paragraph of perspectives at the end of the discussion section.

  1. Please include limitations of the study since the sample size for each group is small and differences about somatic mutation variants were not significant between the two groups.

We acknowledge that the small sample size is one of the main limitations of our study, as we have stated in the Discussion section (lines 356-359). We have also added in the text how the small sample could affect the lack of statistically significant differences between the two groups of patients regarding somatic mutation variants (lines 359-361).

  1. I suggest to include section on possible perspective deriving from these findings. Some references eventually should be added to broaden the discussion:
  • Future Sci OA. 2019 May 3;5(5):FSO394. doi: 10.2144/fsoa-2019-0017. Review about the relationship between cigarette smoke-cell cycle and tumor spread.
  • HIF in tumor progression associated with EMT pathway which is in turn linked to IGF–R Activation
  • Sci Rep. 2013;3:2560. doi: 10.1038/srep02560 about the anti-EGFR resistance due to EMT-IGF crosstalk
  • J Surg Oncol . 2000 Oct;75(2):98-102 about the use of novel biomarkers for aggressive colon cancer on peripheral blood
  • Int J Environ Res Public Health. 2014 Aug 25;11(9):8645-60. About clinical parameters associated with KRAS mutation.

Following the reviewer’s suggestion, we have added a paragraph at the end of the Discussion section focusing on the therapeutic perspectives deriving from our findings (lines 367-379). We have also analyzed the references provided by the reviewer aiming to broaden the discussion.

Reviewer 2 Report

DNA sequencing of Blood and FFPE samples from Kras wild-type metastatic CRC patients, treated with Cetuximab or Panitumumab, was performed to identify somatic gene variants that predict therapy response. The authors focused on several EGFR-related candidates and identified somatic variants of IGF1R, IRS2, LRIG1, 2, 3, Nras and PDGFRA as putative markers.  

Comments

  1. Several markers such as Her2 amplification predict responsiveness of EGFR-based therapy in Kras wild-type CRC but additional markers are needed. The authors identified somatic variants that might in principle be promising. However, as stated by them in the discussion, the number of analyzed responsive and non-responsive patients is quite small. This undermines the validity of the study and might explain why the authors identified sex as false-positive, predictive marker. Moreover, sidedness of CRC was not predictive although it has been shown that left-sided cancers respond better to EGFR blockade.
  2. The predictive value of identified somatic variants should be reproduced in a second patient cohort.
  3. There are no wet lab data in the manuscript. Ideally, the authors should characterize Kras wild-type CRC organoids for the identified somatic variants and perform EGFR inhibitor studies.

Author Response

  1. Several markers such as Her2 amplification predict responsiveness of EGFR-based therapy in Kras wild-type CRC but additional markers are needed. The authors identified somatic variants that might in principle be promising. However, as stated by them in the discussion, the number of analyzed responsive and non-responsive patients is quite small. This undermines the validity of the study and might explain why the authors identified sex as false-positive, predictive marker. Moreover, sidedness of CRC was not predictive although it has been shown that left-sided cancers respond better to EGFR blockade.

We agree with the reviewer that the small sample size is one of the main limitations of our study we stated in the Discussion section (356-362). As the reviewer highlights, the results regarding the predictive role of sex and sidedness could be attributable to the small sample size. We have now included a sentence in the Discussion section to report this limitation (lines 359-362). Given the small sample size, we have also stated in the Discussion section (lines 365-366) that larger, prospective and functional studies are needed to confirm the validity of our findings.

  1. The predictive value of identified somatic variants should be reproduced in a second patient cohort.

We are aware that the findings obtained in our study should be replicated in a similar and independent cohort of patients to be more conclusive. We have included a sentence in the Discussion section to report this limitation (lines 365-366).

  1. There are no wet lab data in the manuscript. Ideally, the authors should characterize Kras wild-type CRC organoids for the identified somatic variants and perform EGFR inhibitor studies.

We agree with the reviewer that performing functional studies would be of high interest to validate our results. This is one of the limitations of our study, which is mentioned in the Discussion section (line 366).

Reviewer 3 Report

This is a well-design study with some interesting results. The introduction appropriately explains the context, the materials and methods are solid, the results sounds good and the discussion is quite deep.

I have only some minor concerns. Were the mutations detected using NGS also confirmed by a second technique?

Author Response

I have only some minor concerns. Were the mutations detected using NGS also confirmed by a second technique?

To confirm the validity of our results, we also sequenced some of the samples included in the study (P3, P11, P63) with the TruSight Tumor 15 panel (Illumina, San Diego, CA, USA), the panel currently used in our hospital in our daily routine. In this way, we were able to validate the mutations found in RAS, BRAF and PIK3CA genes by our panel. We did not find any discrepancy. Furthermore, all the BRAF V600E mutations previously detected by Sanger sequencing (samples P3, P39 and P55) were again identified by our panel. Together with the review of all the mutations using the Integrative Genomics Viewer, we consider these points guarantee the reliability of our results. We have added a sentence in the Patients and methods section (lines 159-161) to clarify this point and thank the reviewer for pointing this out.

Round 2

Reviewer 2 Report

The authors discussed the reviewers concerns but no additional data have been included. False-positive identification of sex as predictive marker is still a major cause for concern and does not dispel doubts that identified candidate mutations are also false-positives. Data should be reproduced in a second patient cohort.